# Stx2 Induces Differential Gene Expression and Disturbs Circadian Rhythm Genes in the Proximal Tubule

**DOI:** 10.3390/toxins14020069

**Published:** 2022-01-19

**Authors:** Fumiko Obata, Ryo Ozuru, Takahiro Tsuji, Takashi Matsuba, Jun Fujii

**Affiliations:** 1Division of Bacteriology, Department of Microbiology and Immunology, Faculty of Medicine, Tottori University, 86 Nishicho, Yonago 683-8503, Japan; ttj091@gmail.com (T.T.); junfujii@tottori-u.ac.jp (J.F.); 2Department of Microbiology and Immunology, Faculty of Medicine, Fukuoka University, 7-45-1 Nanakuma, Jonan-ku, Fukuoka 814-0180, Japan; ozuru@fukuoka-u.ac.jp; 3Division of Infectious Disease Control and Prevention, Department of Animal Pharmaceutical Science, School of Pharmaceutical Science, Kyusyu University of Health and Welfare, 1714-1 Yoshino-machi, Nobeoka 882-8508, Japan; matsubat@phoenix.ac.jp

**Keywords:** Shiga toxin type 2 (Stx2), renal proximal tubule, mouse, human renal proximal tubular epithelial cell (RPTEC), microarray, circadian rhythm

## Abstract

Shiga toxin-producing *Escherichia coli* (STEC) causes proximal tubular defects in the kidney. However, factors altered by Shiga toxin (Stx) within the proximal tubules are yet to be shown. We determined Stx receptor Gb3 in murine and human kidneys and confirmed the receptor expression in the proximal tubules. Stx2-injected mouse kidney tissues and Stx2-treated human primary renal proximal tubular epithelial cell (RPTEC) were collected and microarray analysis was performed. We compared murine kidney and RPTEC arrays and selected common 58 genes that are differentially expressed vs. control (0 h, no toxin-treated). We found that the most highly expressed gene was GDF15, which may be involved in Stx2-induced weight loss. Genes associated with previously reported Stx2 activities such as src kinase Yes phosphorylation pathway activation, unfolded protein response (UPR) and ribotoxic stress response (RSR) showed differential expressions. Moreover, circadian clock genes were differentially expressed, suggesting Stx2-induced renal circadian rhythm disturbance. Circadian rhythm-regulated proximal tubular Na^+^-glucose transporter SGLT1 (SLC5A1) was down-regulated, indicating proximal tubular functional deterioration, and mice developed glucosuria confirming proximal tubular dysfunction. Stx2 alters gene expression in murine and human proximal tubules through known activities and newly investigated circadian rhythm disturbance, which may result in proximal tubular dysfunctions.

## 1. Introduction

Shiga toxin-producing *Escherichia coli* (STEC) infection causes severe symptom hemolytic uremic syndrome (HUS), which is defined by the triad of acute renal failure, thrombocytopenia, and hemolytic anemia [1]. Shiga toxins (Stxs) are thought to play a major role in damaging cells in this infectious disease. The characteristic renal histopathology of HUS includes glomerular and tubular damages such as capillary fibrin deposition within glomeruli and tubular necrosis, respectively [1,2]. Earlier dysfunction occurrence in the proximal tubules of STEC-infected patients has been reported by analyzing urine. The proximal tubular functional urinary markers N-acetyl glucosaminidase (NAG) and β2 microglobulin (β2MG) from acute stage urine of STEC O157- or O111-associated HUS patients were measured to be highly increased, suggesting early proximal tubular dysfunction [3].

Stxs, including Stx2, consist of a single A subunit and five B subunits (A1B5 toxin), in which A subunit possesses enzymatic activity depurinating adenine 4324 from 28S rRNA of the 60S ribosome subunit, whereas B subunits are needed to bind host cells through globotriaosylceramide (Gb3). There are several known influences that Stxs exert on the cells. (I) By binding to its receptor Gb3, the toxin elicits signal transduction through Src-family kinase Yes [4,5]. After binding to the receptor, Stxs internalize to the cell by endocytosis and are retrogradely transported to the ER. Stxs go through the endosome and the Golgi apparatus where the A subunit is cleaved between catalytic A1 and C-terminal A2 fragments by furin [6], although A1 and A2 fragments are still connected with a disulfide bond. Toxin reaches the ER and the disulfide bond may be reduced; then, an A1 fragment from holotoxin is released [7,8]. This is when influence by (II) ER-stress happens with unfolded protein-mimicking toxin fragments, which induce unfolded protein response (UPR). Both A and B subunits are capable of binding Bip, that is indicative of inducing ER-stress [9,10]. Cleaved A1 fragment is translocated to the cytoplasm and depurinates an adenine from 28S rRNA to (III) inhibit protein synthesis. The modification Stxs cause to 28S rRNA induces (IV) ribotoxic stress response (RSR), which activates downstream signaling cascade [11]. Influences (II) and/or (IV) are known to induce (V) particular gene transcription leading to cytokine secretion [12]. Also, (II) and (IV) are involved in (VI) Stxs-induced apoptosis [13,14,15].

In vitro experiments using human proximal tubules and in vivo mouse models strongly connect proximal tubular damages to Stxs. Both primary cultures and cell lines of human proximal tubules are highly susceptible to Stxs and exhibit cytokine release and cell death, including apoptosis [16,17,18,19,20]. Human primary renal proximal tubular epithelial cells (RPTECs) express the receptor Gb3, are highly susceptible to Shiga toxins, and have been used in many articles to analyze biological reactions such as protein synthesis inhibition, cytokine production and induction of apoptosis [17,20]. Shiga toxin type 1 (Stx1)-injected mice showed glucosuria, which indicates lowered reabsorption of glucose by the proximal tubule [21]. These suggest Stxs toxicity to the proximal tubules both in vitro and in vivo. In STEC-infected mouse models, frequent findings in histopathology are necrosis of tubular epithelial cells and sloughed cells within the lumen of the proximal tubules [22,23,24]. Purified Stx (either Stx1 or Stx2) injected mice also present sloughing of the proximal tubules with relatively intact nuclei [25,26,27].

Though damages of the proximal tubules associated with Stx injection or STEC infection have been described functionally and histologically, gene targets involved in lesioning proximal tubules are yet to be shown.

Here we analyzed gene expressions in Stx2-injected mouse kidneys and Stx2-treated human primary renal proximal tubular epithelial cell (RPTEC) culture to show proximal tubule-associated gene alterations induced by Stx2.

## 2. Results

### 2.1. Stx2-Injected Mouse Kidney Pathology

To identify Stx2-associated murine renal pathology, renal tissue sections from Stx2-injected mice were stained with Periodic Acid-Schiff (PAS). Clear strong pink stain at the luminal side of tubular epithelial cells indicates glycocalyx of brush border, which is characteristic of the proximal tubular cells. We determined cells/nuclei outside of glycocalyx (within the luminal space) as detached from neighboring cells and basement membranes (i.e., sloughed cells). In Figure 1B (8 h post Stx2), C and D (48 h post Stx2), sloughed cells with relatively intact nuclei are shown with arrows. We also observed cell debris within the proximal tubular lumen (Figure 1 arrowheads). After Stx2 administration, these lesions appeared quite clearly, whereas at 0 h (Figure 1A, without Stx2), they were negligible.

### 2.2. Gb3 Expression in Mouse and Human Kidneys

#### 2.2.1. Mouse Renal Cortex Gb3 Expression

As the proximal tubular cells showed histological damages from Stx2 insult, we tested Stx2 receptor globotriaosylceramide (Gb3) expression in murine kidney cortex along with a proximal tubular marker aquaporin-1 (AQP1). As shown in Figure 2, AQP1 positive cells exhibited Gb3 positivity mostly at the luminal side of the cells (cells are marked with closed arrowheads, while lumina are marked with asterisks). We also observed that AQP1 negative tubular Gb3 stains presumably collecting duct cells (Figure 2 open arrowhead) as reported [28]. The intensity of Gb3 stain was stronger in the AQP1 (−) cells, however, the number of this type of tubules was smaller compared to Gb3/AQP1 double-positive proximal tubules in the cortex. CD31 positive endothelial cells are Gb3 negative in the mouse kidney. An isotype-matched antibody control for Gb3 (rat IgM) was used as a negative control. Rat IgM did not react with mouse kidney tissue (Appendix A).

#### 2.2.2. Human Renal Cortex Gb3 Expression

In human renal tissue, Gb3 was also positive in AQP1 positive proximal tubules (Figure 3, closed arrowheads). The localization of Gb3 within the proximal tubular cells was stronger on the luminal side. We also observed AQP1(−)Gb3(+) cells (Figure 3, open arrowheads) as we did in murine tissue. Human endothelial cells express Gb3 (CD31/Gb3 double positive, Figure 3, asterisk). Isotype control sections with rat IgM did not show a positive reaction in human renal tissue (Appendix A).

### 2.3. Microarray Analysis

As both mouse and human proximal tubular epithelial cells are Gb3 positive, it suggests that this cell type is affected directly by Stx2. In order to determine Stx2-affected genes in both murine and human proximal tubular cells, we extracted total RNA from kidneys of Stx2-injected mice and Stx2-treated human primary RPTECs, and analyzed gene expression by microarray.

#### 2.3.1. Differentially Expressed Genes in Both Mouse Kidney and RPTEC

Differentially expressed genes (DEGs), that are common to both mouse kidney and RPTEC were selected. Basis of selecting common DEGs is that (1) genes with log2-fold change (against 0 h) more than 1 or less than −1 in both mouse kidney and RPTEC microarray at least in one time point and (2) distribution of log2-fold change of a gene fits cubic curves rather than quadratic curves to better accommodate gene expression peaks in the middle of time course in the kidney microarray. With the two criteria above, we selected 58 common genes between the Stx2-treated mouse kidney and human RPTEC (Figure 4A). Expression patterns of these 58 genes in the mouse kidney are shown in a heatmap (Figure 4B). Most genes were upregulated at later hours after Stx2 injection; however, some genes had earlier expression patterns. Genes that had more than 1 in log2-fold change were listed for each time point (Table 1).

To analyze functionality of common DEGs, we used the Metascape platform. It assigns Gene Ontology (GO) terms, which consist of molecular function, cellular component, and biological process, to selected genes. We deposited common 58 DEGs to Metascape and obtain top 20 GO terms in which these DEGs belong (Figure 4C). GO terms are listed with corresponding genes in Appendix A. To see a gene list within each GO terms shown in Figure 4C, please refer to Appendix A Metascape enrichment. GO terms associated with cell death, inflammation, and signaling cascades were ranked in the top 20 as expected; however, circadian rhythm drew attention as a novel GO terms. Known connections of common genes were shown with STRING analysis (Figure 4D). In this analysis, genes were clustered in six different groups that were accounted for with a strong association of binding, influences (catalysis, activation or inhibition), as well as co-expression of genes.

#### 2.3.2. Trend of Gene Expression

Traces of several gene expressions are shown in graphs with *x*-axis for the time (h) after Stx2 injection and *y*-axis for log2-fold change against 0 h (Figure 5). Thus, *y*-axis value equals 1 means a 2-fold change vs. 0 h expression.

##### The Earliest Responsive Genes

Among all 58 common genes, pyruvate dehydrogenase kinase isozyme 4 (PDK4) was the only gene that was with more than a log2-fold change of 1 at 4 h time point in mouse kidney time scale (Table 1). PDK4 expression is known to be increased by CAAT/enhancer-binding protein beta (CEBPB) [29], which had a small peak at 4 h (log2-fold change equals 0.41) and had a gradual increase until the end of the time course (Figure 5A).

##### Immediate-Early Genes

Immediate-early genes that respond to toxic cellular insult were upregulated in the early hours (6–12 h) after Stx2 injection. Transcription factors such as subunits of transcription factor activator protein-1 (AP-1) (activating transcription factor 3 (ATF3), FOS, JUN and JUNB) and early growth response 1 (EGR1) mRNAs were upregulated as well as growth differentiation factor 15 (GDF15) (Table 1). Especially, GDF15 was the most expressed gene that had log2-fold change equals 3.72 at 12 h. GDF15 expression along with other immediate-early genes is shown in Figure 5B.

##### EDN-1, EGR-1, and PER1

Endothelin 1 (ET-1, coded by EDN1 gene) is a secreted peptide and a potent vasoconstrictor. In our dataset, EDN1 had a small peak at 6 h then a gradual increase with log2-fold change exceeded 1, after 24 h. Transcriptional regulation of EDN-1, EGR-1 and PER1 is reported to be associated (Figure 5H). PER1 is a key circadian gene repressor.

##### Circadian Genes

Circadian-related genes, which were differentially expressed with log2-fold change greater than 1, were clustered in one group using STRING (Figure 4C, purple cluster). Those included a core clock transcription factor PER1, non-core clock genes nuclear receptor subfamily 1 group D member 1 (NR1D1, Rev-Erbα), and nuclear factor interleukin 3 regulated (NFIL3, e4bp4) along with clock-related gene carbon catabolite repression 4-like protein (CCRN4L, nocturnin (NOCT)) (Figure 5I). By using Circadian Expression Profiles Data Base (CircaDB), 14 genes out of 58 common DEGs were matched to mouse kidney microarray database of circadian rhythm (Table 2).

The circadian rhythm is created by clock circadian regulator (CLOCK)-aryl hydrocarbon receptor nuclear translocator like (ARNTL, brain and muscle ARNT-like 1 (BMAL1)) dimer, which acts on expression of PER1 and cryptochrome circadian regulator 1 (CRY1), then PER1-CRY1 dimer suppresses CLOCK-BMAL1 dimer activity, therefore consecutive waves of transcripts for transcription factors (CLOCK and BMAL1) and repressors (PER1 and CRY1) contribute to the time of the day dependent gene expressions to regulate rhythm [30]. Circadian rhythm genes appeared to lose their rhythms after Stx2 injection (Figure 5I,J). In our dataset, initially, PER1-CRY1 and CLOCK-BMAL1 expressions had alternate peaks maintaining rhythms; however, later in the time course, they gradually lost the rhythm and a disturbance in the renal circadian clock is suggested (Figure 5J).

##### Circadian-Regulated Genes

In the kidney, circadian rhythm controls function through gene expression [31]. Circadian rhythm-regulated proximal tubular Na^+^-glucose transporter SGLT1 (solute carrier family 5 member 1 (SLC5A1)) is down-regulated in Stx2-injected kidney (Figure 5K). SGLT1 (SLC5A1) expression is known to be regulated by PER1 [32] which loses its rhythm in our data set (Figure 5J). Urine glucose measurement in two Stx2-injected mice showed that increased excretion of glucose (Table 3) probably reflecting dysfunction in glucose reabsorption in the proximal tubule epithelial cells.

Another proximal tubular factor regulated by the circadian rhythm is Na^+^/H^+^ antiporter (exchanger), NHE3 (solute carrier family 9 member A3 (SLC9A3)) [32]. NHE3 anchor protein NHE regulator cofactor 1 (NHERF1, SLC9A3 regulator 1 (SLC9A3R1)) connects NHE3 to ezrin-actin, and this gene was down-regulated (Figure 5K).

##### Matricellular Protein Genes

Cysteine-rich angiogenic inducer 61 (CYR61, cellular communication network factor 1 (CCN1)) and connective tissue growth factor (CTGF, CCN2) are matricellular proteins that associate with multifunction such as adhesion, cell proliferation, extracellular matrix (ECM) production, and differentiation. These two genes appeared in many GO functions such as positive regulation of cell death, TNF signaling pathway, response to steroid hormone, response to oxygen levels, regulation of mitogen-activated protein kinase (MAPK) cascade, fluid shear stress and atherosclerosis, p38MAPK cascade, response to peptide, cellular response to growth factor stimulus, response to chemokine, and response to wounding (Figure 4C). They are increased with small peaks at 6 h after Stx2, followed by other peaks at relatively earlier hours of 12 and 24 h, respectively (Figure 5C).

##### UPR Genes

Stxs are known to induce endoplasmic reticulum (ER)-stress or unfolded protein response (UPR) by mimicking unfolded protein in the ER [9,10,13]. One of the hallmarks of UPR is upregulation of CAAT/enhancer-binding protein homologous protein (CHOP, DNA damage inducible transcript 3 (DDIT3), growth arrest and DNA damage-inducible protein 153 (GADD153)) gene transcription and translation. In our dataset, indeed, CHOP transcription was increased after Stx2 treatment (Figure 5D–G).

##### NF-κB Pathway-Associated Genes

A classical/canonical NF-κB pathway inhibitor NFKBIA (IκBα) and an alternative/noncanonical pathway NF-κB subunit RelB were differentially expressed in our data set (Figure 5L). Corresponding to this, NF-κB-regulated inflammatory genes (CCL20, CXCL1 and CXCL10) were also differentially expressed (Figure 5L).

## 3. Discussion

### 3.1. The Proximal Tubular Pathology in STEC Patients and Animal Models

Oliguria is a sign of acute tubular necrosis (ATN). Because of ATN, sloughed epithelial cells obstruct the lumen of tubules to reduce fluid flow that appears as oliguria. In STEC patients, oliguria and anuria are often findings that suggest ATN. Reduction in the proximal tubule function during earlier stages of STEC infectious disease has been shown as increases in urinary β2MG and NAG [3,33]. Β2MG is a 12 kDa small protein that is freely filtered through glomeruli and completely absorbed by the proximal tubules in a healthy state. However, once the proximal tubules are damaged, the amount of β2MG in urine increases thus serves as a biomarker for proximal tubular function. NAG is a large lysosomal enzyme of 130 kDa and the major glucosidase in the proximal tubule. Due to the large size, the glomerulus cannot filter NAG whereas damaged proximal tubules secrete NAG in the urine. Thus, an increase in urinary NAG is another biomarker for proximal tubular damage. Glomerular histopathology such as fibrin-obstructed capillaries is a frequent finding of STEC patients; however, the above urinary data suggests proximal tubular damage in the earlier disease. Also, urine from STEC patients contains casts consisting of tubular epithelium which is a result of the sloughing of damaged proximal tubular cells [1]. In animal models, sloughing of the proximal tubular cells is often found both in STEC infection and Stx injection models [25,26,27,34]. In the current study, we found sloughing of the proximal tubular cells in Stx2-injected murine kidneys suggesting Stx2-induced proximal tubular alteration (Figure 1). Also, we detected Stx2 receptor Gb3 in both mouse and human kidney proximal tubules with luminal side expression that is indicative of a direct interaction of Stx2 with the proximal tubules (Figure 2 and Figure 3). PDK4 phosphorylates pyruvate dehydrogenase to inhibit its activity resulting in a decrease in glucose utilization and an increase in fat metabolism in response to prolonged fasting and starvation. It protects detached epithelial cells against detachment-induced cell death (anoikis) [35]. PDK4 initially had a peak at 4 h, then another peak at 8 h (Figure 5A). It was reduced at 12 h, but from there it had a gradual increase until 72 h with log2-fold change ended up with 3 (12-fold change) (Figure 5A). The detachment of epithelial cells occurred in Stx2-injected mouse proximal tubular cells (sloughing) with relatively intact morphology (Figure 1), this may be reflective of PDK4 upregulation.

As NHERF1 (SLC9A3R1) contributes to the normal cell adherence to the extracellular matrix [36], actin cytoskeleton organization, and maintaining cell morphology, lowered expression of this gene may have consequences such as sloughing of cells (Figure 5K).

Effect of Stx2 on the proximal tubules seems such an important factor for STEC infectious disease; however, alterations in gene expression focusing on in vivo proximal tubular cells were not available before. Keepers et al. [37] showed differentially expressed genes (DEGs) in the kidneys of Stx2+LPS-injected mice such that most of the earlier and largely altered genes were from LPS influence and genes that were altered very late are from Stx2 effect. Their data showed valuable insight into the differential gene expressions of the kidney but did not separate types of cells that were responsible for that change. In the current study, we tried to seek out DEGs that are associated with the proximal tubular cells by comparing microarray data from mouse kidney and human primary proximal tubular cells that were treated with Stx2.

### 3.2. Proximal Tubular Factors Involved in Stx2-Induced Pathology in Relation to Known Stx2 Activities in Gb3 Expressing Cells

#### 3.2.1. Activity (I), Signal Transduction Induced by Stxs Binding to Gb3

B subunits of Stxs induce non-receptor src tyrosine kinase Yes phosphorylation [4]. Phosphorylated Yes activates downstream signaling molecules by its kinase activity and leads to cytoskeleton remodeling [5]. Yes-associated protein (YAP) phosphorylation in Y357 by Yes is known to induce YAP nuclear translocation where it acts as a transcriptional co-activator to induce target genes such as CTGF [38]. Matricellular proteins CYR61 (CCN1) and CTGF (CCN2) are secreted to the extracellular matrix (ECM) and contribute to maintaining cell-ECM adhesion by binding receptor integrins and cell surface-attached heparan sulfate proteoglycans (HSPGs) [39,40]. The key molecule of CCN1 and 2 transcription is YAP, and it is kept inactive (phosphorylated at serine residue 127) in a normal state under the Hippo signaling pathway. Once low cell-cell adhesion is detected, YAP (S127) is no longer phosphorylated allowing it to enter the nucleus, and transcription of CCN1 and 2 increases [41,42,43]. CCN1 and 2 act as wound healing molecules to repair cell lost tissue [38,44]. In our dataset, CCN1 and 2 had small peaks at an earlier time point (6 h). This may be reflecting Stx2 binding-induced Yes activation. Later hours, sloughing may have triggered further YAP activation by switching off the Hippo signaling pathway to induce greater expression of CCN1 and 2 at 12 h or later (Figure 5C).

#### 3.2.2. Activity (II), Inducing ER-Stress/UPR

Stxs reach the ER, where cleaved A1 fragment and B subunits mimic unfolded proteins [9,10,45,46]. Glucose-regulated protein 78 kDa (Grp78, binding immunoglobulin protein (Bip)) is anchoring three UPR key proteins Inositol-requiring enzyme-1 (Ire-1), protein kinase R-like endoplasmic reticulum kinase (PERK) and activating transcription factor 6 (ATF6) in ER membrane during normal state. However, due to higher affinity of Grp78 to unfolded proteins, it releases those anchored proteins. ATF6 is activated after the release and increases CHOP (DDIT3) and X-box binding protein 1 (XBP-1) mRNAs, whereas Ire-1 splices XBP-1 mRNA to produce XBP-1 protein, which contributes to CHOP (DDIT3) transcription. PERK activation leads to ATF4 activation that also increases CHOP (DDIT3) mRNA. Therefore, an increase in CHOP (DDIT3) mRNA is a hallmark of UPR [47,48]. Stxs are reported to induce UPR and CHOP (DDIT3) is upregulated in our dataset (Figure 5D–G).

GDF15 belongs to the transforming growth factor beta (TGFβ) family. Recently, GDF15 is shown to mediate diabetic drug metformin-induced weight loss [49]. In metformin orally administered mice, intestinal and renal GDF15 mRNA was increased resulting in an elevation in serum GDF15. Circulating GDF15 works via a brainstem-restricted receptor to suppress food intake and reduce body weight [50]. In the metformin administered mice, increased renal GDF15 is at least in part under the regulation of CHOP (DDIT3). In our dataset of Stx2-injected mice kidneys, CHOP (DDIT3) increased initially at 4 h with a very tiny peak and then started to have an increasing trend toward 48 h where it maintained the level of a log2-fold change around 1.3 (Figure 5E). According to STRING cluster analysis, GDF15 receives active input from transcription factors CHOP (DDIT3) and ATF3 (Figure 5E) [51,52]. The Stx2-injected murine weight started to decrease or deviate from saline control after 24 h (Appendix A). An increase in GDF15 mRNA in the kidney at 12 h may have influenced the brainstem to reduce food intake, which in turn appeared as a weight loss (Figure 5E and Appendix A). As GDF15 expression level continues to be high after 24 h, mice continued to have lost weight (Figure 5E and Appendix A).

Upon ER-stress, PERK is released from Grp78 (Bip) and oligomerizes to phosphorylate and inhibit eIF2α leading to global translation repression, except for a few UPR responsive proteins such as ATF4, CHOP, and Grp78. GADD34 (protein phosphatase 1 regulatory subunit 15A (PPP1R15A)) is one of the increased proteins under phosphorylated eIF2α (eIF2α(P))-associated inhibition and upregulated in our dataset (Figure 5D,G). GADD34 is a protein phosphatase-1 that dephosphorylates eIF2α(P) to negatively regulate UPR-induced protein synthesis inhibition [53]. GADD34 (PPP1R15A) expression becomes log2-fold change larger than 1 after 12 h of Stx2 in our dataset and continues to increase until 24 h when it plateaus (log2-fold change equals 2.1) and maintains this level until the end (Figure 5D,G). This suggests that Stx2-induced UPR had negative feedback after 12 h. Increased transcription and translation of CHOP (DDIT3, GADD153) is a hallmark of ER-stress and the differential expression pattern resembles GADD34 except log2 larger than 1 occurs after 24 h and gradually plateaus thereafter (log2 equals 1.3) (Figure 5D,G). Also, CEBPB, the predominant dimerizing partner of CHOP, is upregulated in our dataset (Figure 5D,G). These results suggest that while Stx2-induced ER-stress starts from very early in the time course, it increases the effect up to 24 h and maintains this level of ER-stress until the end while a negative feedback arm also balances itself simultaneously.

#### 3.2.3. Activity (III), Ribosome Inactivating Protein (RIP) Activity

By cleaving an adenine from rRNA, Stxs inhibit de novo protein synthesis. Depurinated rRNA influences alteration of ribosomal morphology, which activates signal transduction [11] known as ribotoxic stress response (RSR), leading to characteristic gene activation (see next paragraph).

#### 3.2.4. Activity (IV), Ribotoxic Stress Response (RSR) Activity

ZAK (a MAPKKK) is the known kinase in RSR and activates downstream JUN N-terminal kinase (JNK) and p38 MAPK pathways [12,15]. Stx1 induces JUN and FOS mRNA expression in human intestinal epithelial cells and this event required RIP activity positive Stx1 that it is thought to be under RSR control [15]. Another RSR downstream molecule, p38 MAPK, is known to induce early response genes such as JUN, FOS, EGR1, MAFF, DDIT3 and ATF3 [54], all of which were differentially expressed in our dataset. JNK and p38 MAPK pathways are also known to be activated downstream events of Ire-1 under UPR (Figure 4B,D and Figure 5B,E,F) [47]. Due to this, Stxs may activate these two pathways through RSR and/or UPR. Extracellular signal-regulated kinase (ERK) 1/2 pathway is also shown to be downstream of RSR [55] and DUSP1 (MKP1) gene is known to be induced by the ERK1/2 pathway [56]. Stx1 and anisomycin (another protein synthesis inhibitor) are capable of inducing DUSP1 expression in Caco-2 cells [57]. In our dataset, DUSP1 expression was induced and this may indicate ERK1/2 activation (Figure 4B,D).

#### 3.2.5. Activity (V), Particular Gene Transcription Leading to Cytokine Secretion

IL-8 secretion in Stxs-treated cells, human intestinal cells in particular, has been reported extensively as a downstream factor of RSR [12,58]. Cytokines and chemokines in animal models revealed CXCL1 and CXCL10 expression within Stx2-treated mouse kidneys [37,59]. IL-8 (CXCL8) and CXCL1 are both neutrophil chemoattractants that utilize CXCR2 as their receptor. IL-8 is expressed in humans but not in mice. In our dataset, human RPTEC showed an increase in IL-8 expression by log2 equals 4.6-fold maximum (log2 values were 4.6, 3.52 and 4.48 at 6, 13, and 19 h after Stx2, respectively). As we extracted common DEGs from mouse kidney and human proximal tubular cells that IL-8 was not included in the final dataset. Instead, the molecule with similar function as IL-8, CXCL1 showed a significant increase as a common DEG (Figure 5L). A downstream pathway of UPR is the NF-κB signaling pathway. NFKBIA (IκBα) is an inhibitory factor of the classical (canonical) NF-κB pathway that binds with NF-κB subunits RelA (p65) and p50. Upon phosphorylation by an upstream kinase, NF-kappa B kinase (IKK), NFKBIA (IκBα) is degraded, thus NF-κB can translocate to the nucleus and induce transcription of inflammatory factors. RelB proto-oncogene nuclear factor-kappa B subunit (RelB) is involved in the alternative (non-canonical) NF-κB pathway and induces inflammatory factor expression. RelB-p50 or -p52 dimers act as transcription factor (NF-κB) when phosphorylation of p100 of RelB-p50-p100 or RelB-p100 complexes is done by IKKα homodimers and this step does not involve IκBα. In our dataset, both NFKBIA and RelB were differentially expressed reaching to log2-fold change >1 after 24 and 48 h, respectively (Figure 5F). This suggests that genes under both NF-κB pathways, IκBα (classical) and RelB (alternative), are induced. As CHOP (DDIT3) differential expression exceeds log2 larger than 1 after 24 h, mouse kidney undergoes UPR, thus activated PERK phosphorylates eukaryotic translation initiation factor 2 subunit alpha (eIF2α) to inhibit translation of NFKBIA [60]. For this reason, an increase in NFKBIA mRNA may not inhibit much of the NF-κB pathway, thus transcription of inflammatory factors is allowed (Figure 5L).

Sobbe et al. [61] showed that both the classical (RelA-p50) and alternative (RelB-p52) NF-κB pathways were active in Stx-injected mice detecting upregulation of RelA-target cytokines CCL20, CXCL1 and CXCL10 that were also differentially expressed in our dataset (Figure 5L).

#### 3.2.6. Activity (VI), Stxs-Induced Apoptosis

Cleaved caspase 3 has been found in Stxs-associated cell death [13,15,62]. It has been shown that caspase-3 activation or induction of apoptosis by Stxs occurs in the context of RSR and UPR. In Stxs-induced RSR, both p38 MAPK and JNK pathways are involved [15] as well as ERK activation is reported [55]. UPR on the other hand is known to induce NF-κB, JNK and p38 MAPK pathways. CHOP (DDIT3) under UPR is shown to be involved in apoptosis [63] and it has also been shown in Stxs-associated apoptosis [13]. JNK involvement in UPR-induced apoptosis has been reported as well [64]. In our dataset, several DEGs such as CHOP (DDIT3), NFKBIA, FOS, JUN, and JUNB are involved in RSR and UPR (Figure 5B,F).

### 3.3. Circadian Dysregulation by Stx2

An increase in serum endothelin-1 (ET-1, EDN1) occurs in STEC-HUS patients [65]. Also, experimentally, Stx2 induces EDN1 expression by activating signal transduction cascades involving MAPKs [66], while UPR is known to induce EDN1 transcription [67]. In our dataset, Stx2 treatment-induced EDN1 expression in both mouse kidney and human proximal tubular cells (Figure 5G). As EDN1 gene expression was a common denominator of mouse kidney and human proximal tubular epithelial cells, it shows EDN1 expression occurs, at least in part, in the proximal tubular epithelial cells in vivo in addition to endothelial cells. Also, secreted ET-1 induces EGR1 expression via MAPKs activation [68]. Our data confirms upregulation of EGR1 in timing when EDN1 is upregulated (Figure 5H). EGR1 induces circadian rhythm key factor PER1 expression [69], thus, adjusts the amplitude of other circadian factors (Figure 5H).

In the circadian regulation, heterodimer of CLOCK-BAML1 (ARNTL) upregulates PER1 and CRY1. In return, PER1-CRY1 heterodimer binds to CLOCK-BAML1 (ARNTL) to suppress further transcription of their own, hence making a negative loop of the circadian rhythm. Thus, expression peaks of positive regulator genes (CLOCK and BAML1 (ARNTL)) and negative regulator genes (PER1 and CRY1) happen in an alternate hour that when CLOCK and BAML1 (ARNTL) have an increase in expression, PER1 and CRY1 are decreased, and the pattern switches in following hours. When we added log2-fold change values of CLOCK and BAML1 (ARNTL) to the graph (Figure 5J), during the earlier time points, PER1 and BAML1 (ARNTL) showed an opposite rhythmic pattern with sharp/narrow peaks, however, after 24 h the peaks started to become dull and continuously increase without rhythm. This implies that Stx2 influenced renal circadian rhythm including within the proximal tubules. Although a further confirmational experiment is needed, our data combined with published literature that describes PER1 effect on renal circadian regulation upon stimuli [70,71], suggest possible Stx2 involvement in circadian rhythm disturbance and effects on proximal tubular functions by reducing renal clock genes-regulated factors such as SGLT1 (SLC5A1) (Figure 5K). Indeed, we observed glucosuria in Stx2-injected mice that reproduces other study with Stx1-injected mice [21] indicating SGLT1 (SLC5A1) dysfunction in the proximal tubules (Table 3). To the best of our knowledge, this is the first report indicating Stx2 involvement in the renal circadian rhythm disturbance.

## 4. Conclusions

We determined important genes concerning proximal tubule-associated expressions that were altered by Stx2. Differential expression of these genes is assumed to be the result of Stx2 activities such as UPR induction, rRNA depurination-associated RSR, and signal transduction induced by plasma membrane receptor Gb3 binding. The most highly expressed gene, GDF15, may influence weight loss caused by Stx2 through brain-stem restricted receptors by reducing food intake. Extracellular matrix and cell adhesion may be influenced by PDK4, NHERF1 (SLC9A3R1), CYR61 and CTGF that cause Stx2-specific histopathology, sloughing. Moreover, disturbance in the renal circadian rhythm by Stx2 may result in proximal tubular functional defects such as glucose reabsorption.

## 5. Materials and Methods

### 5.1. Animals

Mice (C57BL/6, male, 19–22 g, specific pathogen-free) were purchased from Charles River Laboratories Japan, Inc. (Yokohama, Japan). Food and water were given ad libitum. Mice were maintained with a standard 12 h: 12 h light-dark cycle in which light is on at 7 a.m. and off at 7 p.m. Animal room temperature was kept around 24 °C. All procedures were approved by the University Animal Care Committee.

### 5.2. Human Kidney Tissue

Cadaver tissue without any personal identifiers was used in this study, and it is considered exempt by the human investigation guidelines of the University.

### 5.3. Cell Culture

Human renal proximal tubular epithelial cell (RPTEC) primary culture was purchased from Clonetics (Walkersville, MD, USA) and maintained at 37 °C, 5% CO_2_ atmosphere. Cells were cultured in renal epithelial cell growth medium supplemented with human epidermal growth factor, hydrocortisone, epinephrine, insulin, tri-iodothyronine, transferrin, GA-1000 and FBS.

### 5.4. Purification of LPS-Free Stx2

LPS removal from Stx2 was performed as described previously [72]. Briefly, anti-Stx2 11E10 antibody-conjugated column was used to obtain the Stx2 fraction, and the fraction was subsequently passed through Detoxi-gel (Thermo Fisher Scientific, Rockford, IL, USA) to remove LPS. Stx2 fraction was filtered with a 0.2 µm filter and LPS level tested with Limulus amebocyte lysate assay (Pyrotell, Associates of Cape Cod Incorporated, East Falmouth, MA, USA). The Stx2 fraction was determined as having less than 0.03 endotoxin unit (EU)/mL. The Stx2 variant type was Stx2a.

### 5.5. Stx2 Administration to Mice

A 2LD50 dose (5 ng Stx2/20 g body weight), which kills mice within four days (Appendix A), was injected to mice intraperitoneally in a volume of 0.1 mL in LPS-free saline (Otsuka pharmaceutical, Japan). Toxin injection was done between 7 and 9 a.m., in which longer time course mice received Stx2 at 7 a.m. and followed by 24, 48 and 72 h sampling at 7 a.m.

### 5.6. Urine Collection and Urine Glucose Measurement

After 2, 6, 8, 12, 24, 36, 48, 60, and 84 h of Stx2 injection, urine was taken and 10 µL was dropped on to Fuji dry chem slide GLU-P III (FUJIFILM, Kanagawa, Japan) and urine glucose was measured with Fuji Dry Chem 7000 V (FUJIFILM) at the facility managed by. Urine from pre-Stx2 injection was used as 0 h sample. Urine collection was performed with two mice.

### 5.7. Tissue Processing

After 2, 4, 6, 8, 12, 24, 48, and 72 h of Stx2 injection, three mice per time point were euthanized by CO_2_, and kidneys were collected. Half of one kidney was fixed with 4% paraformaldehyde/phosphate-buffered saline for 7 days at 4 °C and processed for the paraffin section. Another half of kidney was used to extract RNA for microarray analysis. Three mice without any treatment were used as normal (naïve or 0 h) control. Three mice were injected with PBS (vehicle) and sacrificed at 72 h and served as vehicle control to show there are negligible changes in gene expressions compare to normal control.

### 5.8. Periodic Acid-Schiff (PAS) Stain

Paraffin sections were hydrated and stained with 0.5% periodic acid solution, followed by a rinse with distilled water. Sections were transferred to Schiff reagent. After sections were washed with 0.6% sodium bisulfite solution thoroughly, they were counterstained with hematoxylin.

### 5.9. Stx2 Treatment in Human RPTEC

RPTEC cells were treated with 10 ng/mL of Stx2 for 6, 13 and 19 h under 37 °C, 5% CO_2_. RPTEC without Stx2 treatment was used as a control. Two biological replicas were performed per time point. After washing with PBS, cells were used for RNA extraction.

### 5.10. Free-Floating Immunofluorescence Staining

For free-floating section processing, a normal mouse was perfused-fixed as described in Obata et al. [69]. Murine kidneys were then dissected and immersed in 4% PFA/PBS for 3 h at 4 °C with gentle agitation. Tissues were washed with PBS, cryoprotected with 30% sucrose/PBS at 4 °C overnight or until sinking to the bottom. Thick frozen sections (50 μm) were cut with sliding microtome with frozen settings. Sections were blocked with 1:50 goat anti-rat IgM antibody in PBS (F104UN, American Qualex International, Inc., San Clemente, CA, USA) for 1 h and incubated with anti-Gb3/CD77 antibody 1:100 in the blocking solution or isotype control (rat IgM) in a matched concentration as the primary antibody for overnight at 4 °C. After washing, sections were incubated with a secondary antibody (anti-rat IgM Alexa Fluor 488, 1:2000 dilution in PBS) for 2 h at 4 °C. Sections were stained for other probes AQP1 (1:1000 dilution, Sigma-Aldrich Japan, Tokyo, Japan) and CD31 (550274, 1:50 dilution, BD Biosciences, Franklin Lakes, NJ, USA) with secondary antibodies anti-rabbit IgG Alexa Fluor 546 (1:2000 dilution in PBS) and anti-mouse IgG Alexa Fluor 647 (1:2000 dilution in PBS), respectively. After washing with PBS, 4′,6-Diamidino-2-phenylindole (DAPI, D21490, Thermo Fisher) stain was done for 15 min at 4 °C. Sections were immersed in fluids in a well of the 96-well plate (U-shaped) with a tiny painting blush each time the solution was changed. The washing step was done in larger wells such as in 12- or 6-well plates. Aqueous mounting media was used for coverslipping. For human tissue preparation, postmortem kidney was fixed in 20% formalin for 2 weeks and processed for free-floating preparation as done for mouse tissue with one exception that anti-human CD31 antibody (CBL468, 1:20 dilution, Merck KGaA, Darmstadt, Germany) was used.

### 5.11. Microarray Analysis

Half of the kidney is immersed in 2 mL of RNALater (Ambion, Austin, TX, USA) at 4 °C and total RNA was extracted with the Rneasy Midi kit (Qiagen, Santa Clarita, CA, USA). Mouse Genome 430A 2.0 arrays (Affymetrix, Santa Clara, CA, USA) were used. CEL files were obtained and normalized with robust multi-chip averaging (RMA), calculation of RNA copy numbers and fold changes against 0 h were performed in R (v.3.6.0) with its package affy [73] and RankProd 2.0 [74]. Data is shown in log2-fold change values. The dataset is deposited to Gene Expression Omnibus (GEO) (Accession number GSE172465).

From RPTEC cells, total RNA was extracted with RNAgents Total RNA Isolation System (Promega, Tokyo, Japan). Whole human genome oligonucleotide microarray (44K oligonucleotide DNA microarray, Agilent Technologies, Tokyo, Japan) was used for microarray experiments. The total RNA samples were used for the preparation of Cy5- and Cy3-labeled cDNA probes. Fluorophore-labeled samples were hybridized on each glass slide and washed, then scanned with a DNA microarray scanner (model G2505A; Agilent Technologies). The Feature Extraction and Image Analysis software (Agilent Technologies) was used to locate and delineate every spot in the array and to integrate the intensities, which were then filtered and normalized by using the locally weighted scatterplot smoothing method. The reproducibility of microarray analysis was assessed by two repetitions of dye-swap in each experiment. Differences in mRNA expression between control and Stx2-treated RPTEC harvested at 6, 10, or 19 h were compared. TXT format data from GeneSpring software was used to determine gene name from probe ID and convert fold changes to log2-fold change. The dataset is deposited to GEO (Accession number GSE172466).

### 5.12. Bioinformatic Analysis of Microarray Data and Visualization

In both mouse kidney and RPTEC microarray data, genes that are common to both and where the change in expression was more than 2-fold decrease or increase (or log2-fold change is either smaller than −1 or larger than 1) at any time point were selected by R. Moreover, in mouse kidney microarray data. Gene alteration that fit cubic curves were chosen using R package maSigPro [https://www.bioconductor.org/packages/release/bioc/html/maSigPro.html (last accessed on 4 January 2022)] [75]. The cubic curve is suitable to accommodate those genes’ increase or decrease during a time course. A higher polynomial degree such as a cubic curve allows to select genes with multiple alterations during the time course better than a quadratic curve. Genes that behaved by similar trajectory were clustered and their decrease/increase patterns were visualized by a heat map using R package gplots [https://CRAN.R-project.org/package=gplots (last accessed on 4 January 2022)] [76]. Gene ontology (GO) was assigned and functions of GO among common genes were enriched using Metascape [http://metascape.org (last accessed on 4 January 2022)] [77]. Connection/relation of common genes were analyzed using STRING [https://string-db.org/ (last accessed on 4 January 2022)] [78]. Association between common genes were searched manually as well as using the textmining function of STRING.

## Figures and Tables

**Figure 1 toxins-14-00069-f001:**
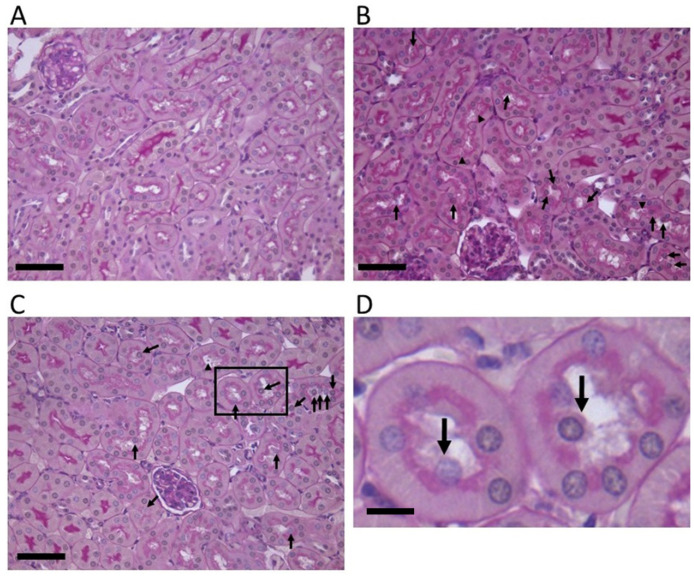
Stx2 induces the sloughing of proximal tubular cells in the murine kidney. Periodic acid-Schiff (PAS) stains of Stx2-injected murine kidney sections are shown. (**A**) 0 h (naïve), (**B**) 8 h, and (**C**) 48 h after Stx2 injection. Arrows indicate sloughing proximal tubular cells. Arrowheads indicate cell debris. A magnified field within (**C**) is shown in (**D**) with sloughing cells pointed with arrows. Bars in (**A**–**C**) indicate 50 µm. The scale bar in (**D**) indicates 10 μm.

**Figure 2 toxins-14-00069-f002:**
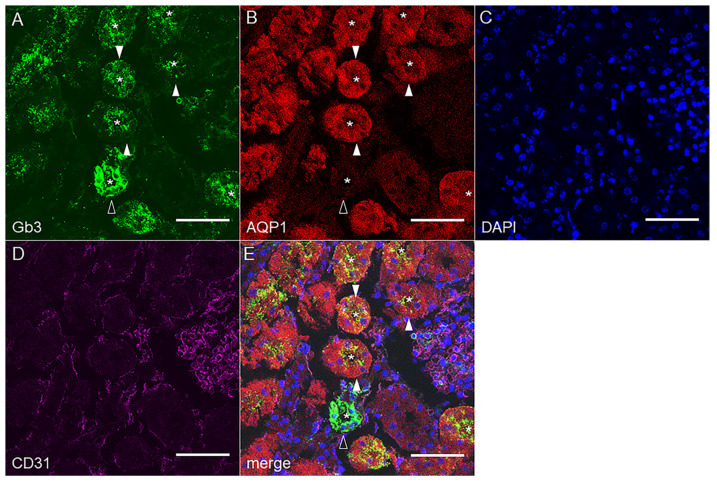
Murine proximal tubules express Gb3 in vivo. Confocal microscopic observation of (**A**) anti-Gb3, (**B**) anti-AQP1, (**C**) DAPI, and (**D**) anti-CD31 immunofluorescence staining. (**E**) A merged picture of A through D. Closed arrowheads show AQP1 positive proximal tubules which are Gb3 positive. An open arrowhead points to AQP1(−)Gb3(+) tubules. Asterisks indicate tubular lumen. Bars indicate 50 µm.

**Figure 3 toxins-14-00069-f003:**
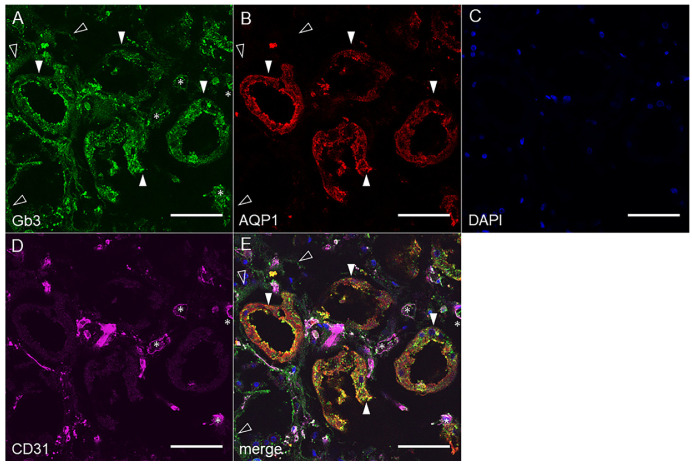
Human proximal tubules express Gb3 in vivo. Confocal microscopic observation of (**A**) anti-Gb3, (**B**) anti-AQP1, (**C**) DAPI, and (**D**) anti-CD31 immunofluorescence staining. (**E**) A merged picture of A through D. Closed arrowheads point to AQP1 positive proximal tubules which are Gb3 positive. Open arrowheads point to AQP1(−)Gb3(+) tubules. Asterisks indicate CD31(+) endothelial cells that are Gb3(+). Bars indicate 50 µm.

**Figure 4 toxins-14-00069-f004:**
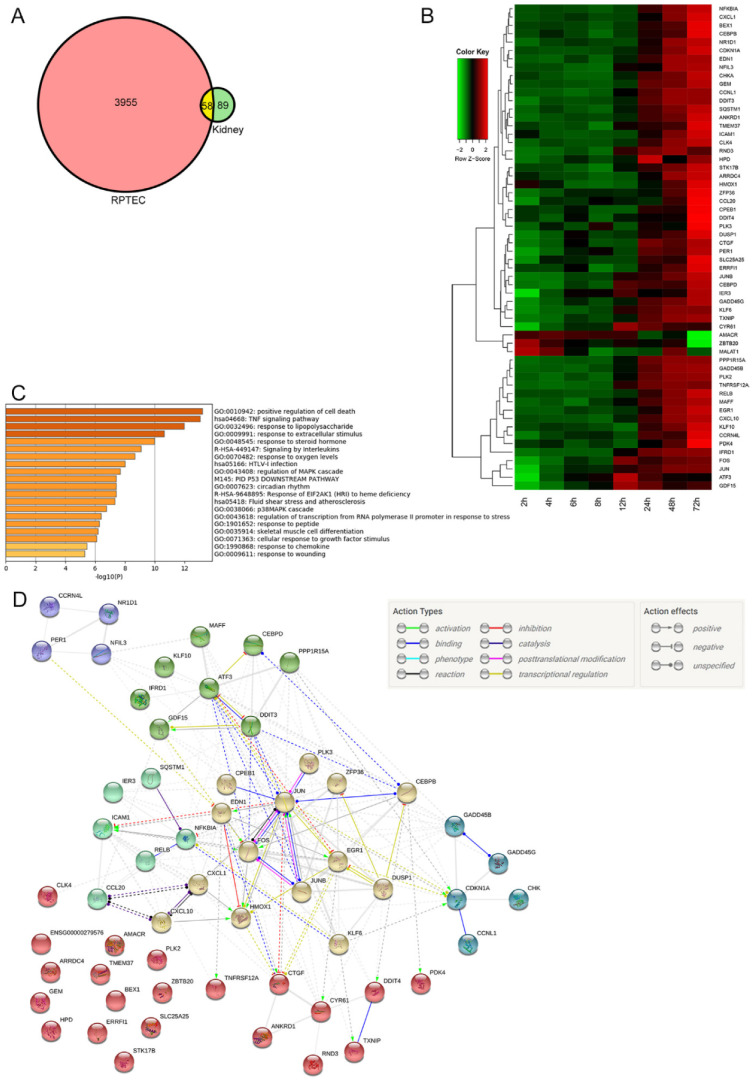
Microarray analysis. (**A**) Among genes analyzed by microarray, 147 genes (green circled), and 4013 genes (red circled) from Stx2-treated mouse kidney and RPTEC, respectively, were detected as having log2-fold change less than −1 or more than 1. Genes that were altered by Stx2 treatment against 0 h in both microarrays (common differentially expressed genes/DEGs) totaled 58 (red and green overlapped area). (**B**) The heatmap of the common 58 genes is shown in mouse kidney sampling timeline. Z equals 0 denotes an average of log2-fold change values of a gene and reflects a trend of gene expression. Thus, greenish color represents a lower value than an average of a particular gene in a row, whereas reddish color represents above the average change. Log2-fold data of the 58 genes from PBS-injected 72 h dataset compared to 0 h is presented as Appendix A for a reference. (**C**) The Metascape GO enrichment bar graph is shown. *X*-axis shows −log10 p values of enriched GO terms across 58 common DEGs that higher number of the enriched GO term is more likely relevant to the system. The top 20 enriched GO terms are shown. (**D**) STRING analysis of 58 common DEGs is shown in the six most connected/related clusters. Spherical colors denote genes belonging to the same cluster. Relations of actions between genes are shown in color- and end shape-coded manner. ENSG00000279576 has now been coded as ENSP00000485396, which denotes human ID of gene MALAT1.

**Figure 5 toxins-14-00069-f005:**
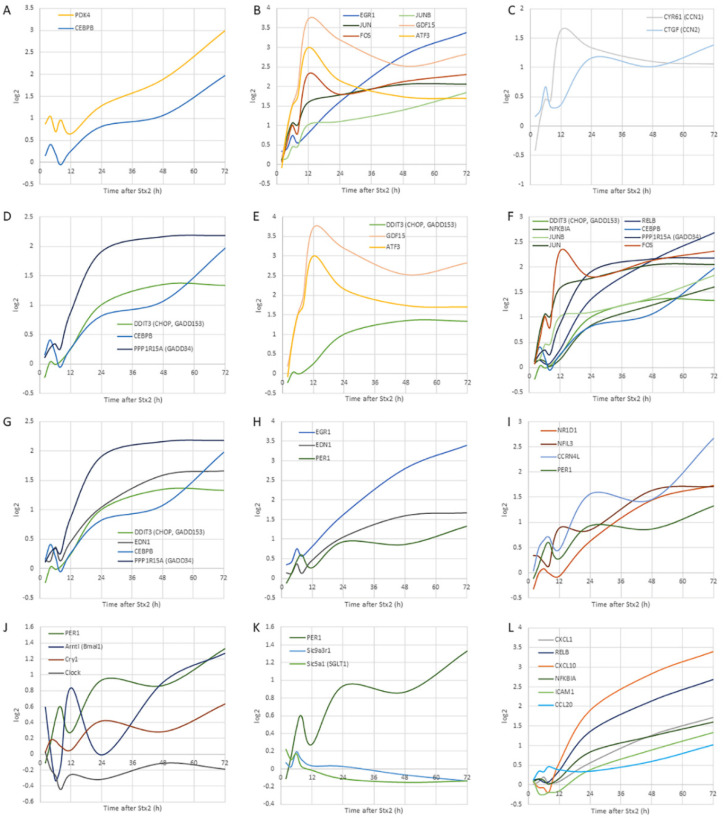
Differentially expressed genes with mouse sampling timeline. (**A**) PDK4 and CEBPB mRNA expression patterns are shown. (**B**) immediate-early genes. (**C**) Matricellular proteins CYR61 (CCN1) and CTGF (CCN2) expressions. (**D**) UPR-related genes DDIT3 (CHOP, GADD153), DDIT3 binding partner CEBPB and negative feedback factor PPP1R15A (GADD34) are shown. (**E**) Transcription factors DDIT3 (CHOP, GADD153) and ATF3 and GDF15, which is transcriptionally regulated by these factors, are shown. (**F**) UPR- and RSR-associated genes are shown. (**G**) EDN1 in relation to UPR genes are plotted. (**H**) Early peaks of EDN1 and EGR1 and following peak of PER1 are shown with later gradual increase in all three genes. (**I**) Genes from Figure 4D purple cluster are circadian rhythm-related genes. (**J**) Circadian core genes PER1, Arntl (BMAL1), Cry1 and Clock are shown. (**K**) PER1 and circadian clock-regulated renal genes are shown. (**L**) Inflammatory genes with NF-kB pathway-related genes are shown. To see individual log2 data points, please refer to Appendix A. Appendix A, showing average log2 with standard deviation bars, is presented as a reference.

**Table 1 toxins-14-00069-t001:** Common genes with log2 larger than 1 at specific time points.

Time after Stx2 (h)	Number of Common Genes log2 > 1	Name of Genes
2	0	n.a.
4	1	PDK4
6	4	ATF3, FOS, GDF15, JUN
8	3	ATF3, GDF15, JUN
12	11	ATF3, CEBPD, CYR61, FOS, GDF15, IER3, IFRD1, JUN, JUNB, KLF6, TNFRSF12A
24	28	ATF3, CCRN4L, CEBPD, CTGF, CXCL10, CYR61, DDIT3, EDN1, EGR1, FOS, GADD45B, GADD45G, GDF15, HPD, IFRD1, JUN, JUNB, KLF10, KLF6, MAFF, PDK4, PLK2, PPP1R15A, RELB, RND3, SLC25A25, TNFRSF12A, TXNIP
48	39	ATF3, CCNL1, CCRN4L, CDKN1A, CEBPB, CEBPD, CHKA, CLK4, CTGF, CXCL1, CXCL10, CYR61, DDIT3, DUSP1, EDN1, EGR1, ERRFI1, FOS, GADD45B, GADD45G, GDF15, GEM, IFRD1, JUN, JUNB, KLF10, KLF6, MAFF, NFIL3, NFKBIA, NR1D1, PDK4, PLK2, PPP1R15A, RELB, RND3, SLC25A25, TNFRSF12A, TXNIP
72	55	ANKRD1, ARRDC4, ATF3, BEX1, CCL20, CCNL1, CCRN4L, CDKN1A, CEBPB, CEBPD, CHKA, CLK4, CPEB1, CTGF, CXCL1, CXCL10, CYR61, DDIT3, DDIT4, DUSP1, EDN1, EGR1, ERRFI1, FOS, GADD45B, GADD45G, GDF15, GEM, HMOX1, HPD, ICAM1, IER3, IFRD1, JUN, JUNB, KLF10, KLF6, MAFF, NFIL3, NFKBIA, NR1D1, PDK4, PLK2, PLK3, PPP1R15A, RELB, RND3, SLC25A25, SQSTM1, STK17B, TMEM37, TNFRSF12A, TXNIP, ZFP36

**Table 2 toxins-14-00069-t002:** Common DEGs matched in CircaDB.

Gene Name	Description	Database
Nr1d1	nuclear receptor subfamily 1, group D, member 1	a
Nfil3	nuclear factor, interleukin 3, regulated	a, b
Cdkn1a	cyclin-dependent kinase inhibitor 1A (P21)	a
Slc25a25	solute carrier family 25 (mitochondrial carrier, phosphate carrier), member 25	a
Dusp1	dual specificity phosphatase 1	a
Chka	choline kinase alpha	a
Ctgf	connective tissue growth factor	b
Ccrn4l	CCR4 carbon catabolite repression 4-like (S. cerevisiae)	a
Per1	period homolog 1 (Drosophila)	a
Amacr	alpha-methylacyl-CoA racemase	a
Edn1	endothelin 1	a
Gadd45g	growth arrest and DNA-damage-inducible 45 gamma	b
Zfp36	zinc finger protein 36	a
Pdk4	pyruvate dehydrogenase kinase, isoenzyme 4	a

(a) Mouse 1.OST Kidney (Affymetrix); (b) Mouse Kidney Rudic 2004 (Affymetrix).

**Table 3 toxins-14-00069-t003:** Urine glucose (mg/dL) after Stx2 injection.

	Time after Stx2 (h)									
Mouse ID	0	2	6	8	12	24	36	48	60	84
1	20	19	30	24	15	20	18	15		94
2	19	20	18	17	20	10	15	43	118	

## Data Availability

Mouse kidney and RPTEC microarray data were deposited to Gene Expression Omnibus (GEO) under Accession numbers GSE172465 and GSE172466, respectively.

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
