# Peer review of "Stx2 Induces Differential Gene Expression and Disturbs Circadian Rhythm Genes in the Proximal Tubule"

_toxins, 2022, doi:10.3390/toxins14020069_

Round 1

Reviewer 1 Report

In the manuscript „Stx2 induces differential gene expression and disturbs circadian rhythm genes in the proximal tubule” the authors show data obtained by injection Stx2 in mouse kidney and intoxication of human primary renal proximal tubular epithelial cell cultures. From those intoxications, microarray analyses over a timeline of several days are generated and the genes which showed log2-fold changes of 1 or -1 in mice and cell culture are described in detail. Some of the identified genes belong to the functional cluster of circadian genes. Changes introduced by Shiga toxin intoxication were not described so far.

This aspect is very interesting. However, in my opinion, there are some aspects and questions regarding the data evaluation and interpretation as well as statements which are not covered in the manuscript.

My major comments are:

  • Reference 9 and 10 don’t support the ER-stress statement in LINE 53 and other statements connected to those publications within the manuscript
  • Results figure 2 and 3 could be one figure.
  • Use the same terms in figure captions and in the text to describe presented data, e.g. LINE 100 white vs. closed arrow heads
  • Figure 4: heat map shown for mice data. Are the changes found in the microarrays for the primary cell line the same and do they show a comparable heat map? Or are there genes which are upregulated in mice and downregulated in the primary cell culture and vice versa?
  • LINE 160:
    • against 0h sample. This indicates that the values obtained in the microarrays at time point zero are used for calculation the changes in gene expression. Regarding the statements and interpretation of the changes in the circadian genes I think this is the wrong reference. I think log changes calculated to PBS treated mice, or cell culture from a same time point would result in much more robust data and will show a clear effect of Stx2 on the expression of those genes.
    • In addition: against 0h is a very important information for the interpretation of the microarray data and should be mentioned in the introduction of the experiment in the text of the manuscript
  • Chapter Trend of Gene expression:
    • Chapter structure is not clear, first genes according to their response time are described and then genes according to functional clusters are described. This makes the understanding rather difficult. I would suggest to stick to one schema of introducing the altered genes.
    • Remove the “subheaders”
    • Include instruction of all functional clustered identified by altered gene expression
  • LINE 241: Scientific consensus does not describe the functionality of Stx as described here. What are the references that support this view?
  • Figure 5: I am not convinced by the way the data is presented.
    • Microarray analyses are not continuous measurements, so, how are the curves generated? I would prefer to see averaged data points with error bars. The presented lines are not supported by the data obtained in the experiment.
    • More detailed presentation of the data is in S4: in this figure huge variances between data points are visible. What is the statistic significance of the data shown here? Are only fold changes with statistical significance included? Please include significance evaluations of the data
  • LINE 328: References do not support statement
  • LINE 365-371: time line seems comparable to cytotoxicity assays. Can the intoxication of the primary cell culture be morphological observed as in HeLa, Vero, or other cell cultures?
  • LINE 457: Which published literature exactly?
  • Methods:
    • Which stx2 variant was used?
    • Which experiment included the human kidney tissue?
    • More details on housing conditions of animals. Temperatures for example can have severe effect on the effect observed by bacterial toxins in animal experiments. So this information is essential.
    • LINE 505: 84 h or 72 h. Why not at 72h?
    • details on statistic evaluation
  • Figure S3: how was significance detected?

Reviewer 2 Report

In this study the authors assessed differential gene expression in murine kidneys and RPTEC in response to Stx2. 58 genes appeared differentially regulated in cells exposed to Stx2  as compared to non-exposed cells. Among those genes the authors found genes previously associated to Stx2 activities and, more surprisingly, circadian clock genes.

This manuscript is well written, data is well presented and it contains interesting data for the community. To strengthen their data the authors might be interested to confirm their micoarray data by RT-qPCR. In addition, although not necessary, the authors might want to validate the data by showing that identified genes are involved in Stx2 response.

As minor points:

In the legend of Fig 2 and Fig 3, the authors mentioned the letters A to E which are shown in the Figure. Should be corrected.

Also there is no asterisk in Figure 3 although it is mentioned in the legend.

In Figure 2: Not clear that Gb3 stains stronger in the luminal side.

Round 2

Reviewer 1 Report

The authors addressed all my questions and comments in their letter in great detail. With the changes/additions in the text and the extra supplement figure I think the authors improved their manuscript. 

With this I think the manuscript is suitable for publication.

This manuscript is a resubmission of an earlier submission. The following is a list of the peer review reports and author responses from that submission.

Round 1

Reviewer 1 Report

The submitted manuscript explores the factors altered by Shiga toxin (Stx) within the proximal tubules. In general, this manuscript shows a correlation between genes concerning proximal tubule (PT) and Stx2. The manuscript explains potential relations between PT and Stx2, but the data doesn’t provide information about other potential factors or confounders. Also, there should be more discussion to differentiate correlation from causation relation between these two phenomena.

The manuscript is written in a sound way and is easy to follow, except the aforementioned coment, there are some questions and concerns about the current format that should be addressed. In the following section,

  • In the title “Stx2 Induces Differential Gene Expression by Activating Several Pathways and Disturbs Circadian Rhythm Genes in the 3 Proximal Tubule”, there is a grammatical error. Also “Activating Several Pathways” doesn’t provide much information as is a known factor. I suggest adding more information or simply drop “Activating Several Pathways” part.
  • Figure 4 should be regenerated to show the sections precisely in one page.
  • The tool or resource that is used to generate figure 4-C should be mentioned. The figure 4-C should be accompanied by a table listing the genes in each GO category (potentially in the supplementary material). That helps in understanding and interpreting of the GO terms.

Reviewer 2 Report

The work “Stx 2 induced differential gene expression by activating several pathways and disturbs circadian rhythm genes in the proximal tubule” investigates changes in gene expression which occur upon direct Stx2 intoxication of kidney cells in both human cells and murine tissue. After initially highlighting the pathology of Stx2 injected mouse tissue, the authors evaluate the presence of Gb3 in murine and human renal cortex. In both tissues Gb3 is detected. This leads to the key question of the manuscript. How does Stx2 intoxication affect gene expression in these two tissues? To investigate this, primary RPTEC and mouse kidneys are treated with Stx2 and the changes in gene expression evaluated by microarray analyses. Common DEGs between murine and human tissue are further evaluated in detail. Besides already described gene, the authors newly identify genes which are associated with the circadian rhythm, which also influence the proximal tubular dysfunction.

The basic concept of the studies seems reasonable and logical. However, several major questions were left unanswered in the manuscript:

  1. How does Stx2 intoxication affects the mouse and human renal cells in their morphology. Are there any differences in the Immunostaining?
  2. Show the RPTECs the same results as the human renal tissue when immunostained for Gb3? Add to supplemental material.
  3. Is the data set obtained for 0 h used as reference to calculate all fold-changes? If yes, in my opinion, is this the wrong reference. Untreated samples must be used as reference to calculate fold-changes and evaluate the effect on gene expression over time. Especially with respect to the time aspect, the reference data sat must be untreated cells, which were cultivated as the Stx-treated cells.
  4. Why was the STRING analysis performed only with 56 of the 58 identified genes? Please clarify (line 142). Why are there 58 DEGs shown (line 132)? Or were two DEGs not available in the STRING database?

Minor comments are:

  1. Abstract: Conclusion is to bold, as it implies a causality between circadian rhythm disturbance and tubular dysfunction, which was not explicitly experimentally shown.
  2. The introduction is quite superficial and should be more adjusted to the results and discussion part
  3. Introduction to results different results and discussion subsections (before lane 57, 74, 237)
  4. Figure 1: Scalebar in D
  5. Figure 3: Target of open arrow unclear, not good visible
  6. Do not use mathematic operators in the text =, >, etc.
  7. Figure 5: legend and figure caption does not fit all the time, different ranges on y-axis makes the data presentation unclear. Include more details in Figure caption. Which DEG is depicted in which panel?

One basic remark: the data presented in figure 5 is based on a few single data points and is not continuously measured. So I would highly recommend to show datapoints and not smoothed lines predicted by Excel.

  1. Include clear cross references in the discussion on the results part.
  2. Several aspects in the discussion belongs in the introduction. E.g., Stx description…
  3. Discuss line 363 – 365 in greater detail. How does the presented data relate to this aspect?
  4. Spacing between words are often odd. One extreme example line 318-324
  5. Line 318: clarify and reference?
  6. Line 386: include IL-8 data in the supplemental data
  7. Line 387-389 clarify
  8. Line 389- 405: NFKBIA vs. IκBα - use one
  9. Line 417: state which DEGs in particular
  10. Why were male mice used? Is a difference based on the sex expected?